# Palladium-Catalyzed *β*-C(sp^3^)–H Bond Arylation of Tertiary Aldehydes Facilitated by 2-Pyridone Ligands

**DOI:** 10.3390/molecules29010259

**Published:** 2024-01-03

**Authors:** Ziting Xu, Zhi Li, Chong Liu, Ke Yang, Haibo Ge

**Affiliations:** 1Jiangsu Key Laboratory of Advanced Catalytic Materials & Technology, School of Petrochemical Engineering, Changzhou University, Changzhou 213164, China; s22020703047@smail.cczu.edu.cn (Z.X.); lz20070303024@126.com (Z.L.); 2Department of Chemistry and Biochemistry, Texas Tech University, Lubbock, TX 79409, USA; chong.liu@ttu.edu

**Keywords:** palladium-catalyzed, 2-pyridone ligand, tertiary aldehydes

## Abstract

2-Pyridone ligand-facilitated palladium-catalyzed direct C–H bond functionalization via the transient directing group strategy has become an attractive topic. Here, we report a Pd-catalyzed direct *β*-C(sp^3^)–H arylation reaction of tertiary aliphatic aldehydes by using an *α*-amino acid as a transient directing group in combination with a 2-pyridone ligand.

## 1. Introduction

Aliphatic aldehydes are not only important intermediates in chemical synthesis, but also ubiquitous structural units in pharmaceuticals and natural products [1,2,3]. Among various synthetic approaches for aliphatic aldehydes, transition metal-catalyzed C–H bond functionalization represents one of the most efficient tools for the construction and derivatization of aliphatic aldehydes [4,5,6,7,8,9]. Recently, the transient directing group strategy (TDGS) has been well applied in the field of transition metal-catalyzed C–H bond functionalization of aldehydes, ketones, and amines [10,11,12,13,14,15]. In 2016, the Yu group reported the Pd-catalyzed C–H arylation of *o*-alkyl benzaldehydes and aliphatic ketones by employing *α*-amino acids as transient directing groups [16]. Meanwhile, our group also disclosed the first example of Pd-catalyzed direct *β*-C(sp^3^)–H arylation of aliphatic aldehydes by using either 3-aminopropanoic acid or 3-amino-3-methylbutanoic acid as a transient directing group [17]. Although much significant progress has been made in this research area, the *β*-C(sp^3^)–H bond functionalization of tertiary aldehydes is rare with only a few examples reported [18,19]. In 2017, the Bull group disclosed Pd-catalyzed *β*-C(sp^3^)–H arylation of tertiary aldehydes with simple *N*-tosylethylenediamine as transient directing group (Figure 1a) [18]. Later, the same group reported the Pd-catalyzed intramolecular *β*-C(sp^3^)–H arylation of tertiary aldehydes in the presence of 2-methoxyethan-1-amine (Figure 1b) [19]. In 2019, the Chen and Zhou group demonstrated Pd-catalyzed selective C–H arylation of phenylacetaldehydes using L-valine as the transient directing group (Figure 1c) [20]. Recently, the Yang and Li group used a calix[4]arene-derived diamine as the transient directing group to achieve this process (Figure 1d) [21]. Unfortunately, these strategies only provided limited examples with isolated yields no more than 63%. Therefore, the development of highly efficient methodologies with a broader substrate scope is desirable.

Recently, the use of a transient directing group in combination with an external 2-pyridone ligand has emerged as a promising strategy in Pd-catalyzed C–H functionalization reactions [22,23,24,25,26,27,28,29,30,31]. It is believed that the 2-pyridone ligand can effectively stabilize the palladium catalyst and lower the transition-state energy of the C–H bond cleavage, promoting and accelerating this catalytic process [22,23,24,25,26,27,28,29,30,31,32,33,34]. The Yu group first found that versatile 2-pyridone ligands could significantly improve the efficiency of TDG-enabled Pd-catalyzed C–H arylation and fluorination of alkyl amines [22,23]. Subsequently, the Yu and Sorensen groups disclosed that 2-pyridone ligands could promote the Pd-catalyzed C–H arylation of aliphatic ketones [24,25,26]. Additionally, Zhang and co-workers reported an effective TDG-enabled Pd-catalyzed *ortho*-C–H chlorination of benzaldehydes with the assistance of external 2-pyridone ligands [27]. Moreover, the Bull group disclosed the Pd-catalyzed methylene C(sp^3^)–H *β*,*β′*-diarylation of cyclohexanecarbaldehydes enabled by a transient directing group and 2-pyridone ligand [28].

In 2020, our group discovered the first example of Pd-catalyzed *γ*-C(sp^3^)–H arylation of aliphatic aldehydes by concurrently using L-phenylalanine as a transient directing group and 3-nitro-5-(trifluoromethyl)pyridin-2-ol as an external ligand [29]. Later, both our and the Yu groups demonstrated that 5-nitro-3-(trifluoromethyl)pyridin-2-ol could promote the palladium-catalyzed direct *β*-C(sp^3^)–H arylation of primary aliphatic aldehydes [30,31]. Encouraged by the above results, we report here a palladium-catalyzed amino acid-enabled direct *β*-C(sp^3^)–H arylation reaction of tertiary aliphatic aldehydes in the presence of a 2-pyridone ligand (Figure 1e).

## 2. Results and Discussion

On the basis of our previous studies, we commenced our investigation on the reaction of pivalaldehyde (**1a**) and methyl 4-iodobenzoate (**2a**) in the presence of catalytic Pd(OAc)_2_ and stoichiometric amounts of AgTFA with L-phenylglycine (**TDG1**, 40 mol%) and 3-nitro-5-(trifluoromethyl)pyridin-2-ol (**L1**, 30 mol%) at 100 °C under a nitrogen atmosphere (Figure 2 and Table 1). After an extensive solvent screening, it was determined that a mixture of HFIP and HOAc in a ratio of 3:1 yielded the desired arylated product **3a-mono** (42% NMR yield) as well as the diarylated product **3a-di** (34% NMR yield) (Table 1, entries 1–6). The subsequent investigation on the amounts of **L1** and **TDG1** revealed that increasing the loading of **L1** (from 40 to 60 mol%) resulted in an enhancement in the yields of **3a-mono** and **3a-di** to 53% and 35%, respectively (Table 1, entries 7–12). Furthermore, no significant improvements were observed when employing alternative Pd catalysts, such as Pd(TFA)_2_, PdCl_2_, and PdBr_2_ (Table 1, entries 13–15). In the absence of the 2-pyridone ligand, both the yields of **3a-mono** and **3a-di** decreased, while no products were obtained in the absence of a transient directing group (Table 1, entries 16–17).

Next, the effect of both transient directing groups and 2-pyridone ligands on this reaction was examined (Figure 3). While the use of *α*-amino acids **TDG1–5** afforded the mono- and di-arylated products in good yields, 2-aminoisobutyric acid (**TDG6**) proved inefficient. Furthermore, *β*-amino acids **TDG7–10** failed to provide any desired products. These results suggest that L-phenylglycine (**TDG1**) was the optimal transient directing group, presumably via formation of a [5,5]-bicyclic palladium species in this protocol. The subsequent screening of 2-pyridone derivatives revealed that 3-nitro-5-(trifluoromethyl)pyridin-2-ol (**L1**) was the optimal ligand, while other 2-pyridone ligands **L2–10** provided only moderate yields.

With the optimized reaction conditions at hand, the substrate scope study of aryl iodides was carried out (Figure 4). The presence of a strong electron-withdrawing (ester, cyano, nitro, and trifluoromethyl) group on the phenyl ring at the *para*- or *meta*-position of iodobenzene was found to be compatible with our current catalytic process, resulting in the isolation of desired mono- and di-arylated products **3a–g** with good overall yields. Notably, the synthetic applicability of this protocol could be further enhanced by facile conversion of these well-tolerated functional groups into other important moieties. As expected, this catalytic system also exhibited compatibility with 5-iodo-2-(trifluoromethyl)pyridine, providing the desired mono-arylated product **3h** in moderate yield. To our delight, natural products-containing aryl iodide derived from complex organic frameworks, including menthol and fenchol, could also be efficiently converted into the desired products **3i–j** with high overall yields. However, the use of **TDG1** and **L1** yielded only moderate yields when employing unsubstituted or electron-donating substituted iodobenzenes as coupling partners, as well as weakly electron-withdrawing iodobenzenes. The utilization of **TDG2** and **L2** resulted in a significant enhancement in the overall yields of **3k** (from 43% to 66%). Furthermore, the desired products **3l–o** were obtained with good yields from iodobenzenes bearing a methoxy, methyl, or halogen group.

Subsequently, the substrate scope study of aliphatic aldehydes was carried out (Figure 5). The *α*-methyl-*α*,*α*-dialkyl acetaldehydes, such as 2-methyl-2-propylpentanal, 2-methyl-2-propylhexanal, and 2-methyl-2-propylheptanal, produced only the mono-arylated products **3p–3r** in good yields. Furthermore, a tertiary aliphatic aldehyde bearing a cyclohexyl group also provided the desired product **3s** in 55% yield. These results indicate that the functionalization of the methyl *β*-C–H bond is predominantly favored over the methylene *β*-C–H bond. Additionally, when *α*,*α*-dimethyl-*α*-aryl acetaldehydes were employed, only mono-arylated products **3t–u** were obtained in moderate to good yields. It is noteworthy that the ether group was also tolerated and both mono- and di-arylated products **3v–w** were isolated with good overall yields. Unfortunately, non-*α*-quaternary aliphatic aldehydes, including cyclohexanal, 2-methylpentaldehyde, and *n*-pentanal, failed in our current catalytic cycle (**3x–z**).

Based on the above results and previous reports [18,19,20,21,22,23,24,25,26,27,28,29,30,31], a plausible catalytic cycle is proposed in Figure 6 [10,11,12,13,14,15,16,17,18,19,20,21,22,23,24,25,26,27,28,29,30,31,32,33,34,35,36,37]. The reversible imine formation of tertiary aliphatic aldehyde **1a** with **TDG1** generates imine intermediate **A**. Next, the coordination of imine species **A**, 2-pyridone ligand **L1** to the Pd(II) catalyst provides cyclic palladium intermediate **B**. Subsequently, 2-pyridone ligand-assisted *β*-C(sp^3^)–H bond activation occurs with intermediate **B** to give rise to [5,5]-bicyclic palladium intermediate **C**. Next, ligand replacement of intermediate **C** with aryl iodide **2,** followed by an oxidative addition process, provides Pd(IV) intermediate **D**. Then, Pd(IV) intermediate **E** is formed through iodine abstraction of intermediate **D** by AgTFA and is further transformed into intermediate **F** via a reductive elimination process. In the presence of 2-pyridone ligand **L1**, metathesis occurs between intermediate **F** and imine intermediate **A**, resulting in the formation of imine intermediate **G** and regeneration of cyclic palladium intermediate **B**. Imine intermediate **G** finally undergoes hydrolysis to generate the desired product **3-mono** and release **TDG1**. Subsequently, the second arylation of **3-mono** in this catalytic cycle produces the di-arylated product **3-di**.

## 3. Materials and Methods

### 3.1. General Information

All the solvents and commercially available reagents were purchased and used directly. Thin layer chromatography (TLC) was performed on EMD precoated plates (silica gel 60 F254, Art 5715, Yantai Jiangyou Silica gel Development Co., LTD, Yantai, China) and visualized by fluorescence quenching under UV light. Column chromatography was performed on EMD Silica Gel 60 (200–300 Mesh, Shanghai Titan Technology Co., Ltd., Shanghai, China) using a forced flow of 0.5–1.0 bar. The ^1^H and ^13^C NMR spectra were obtained on a Bruker AVANCE III–300 or 400 spectrometer (Bruker Corporation, Billerica, Massachusetts, USA). ^1^H NMR data was reported as: chemical shift (δ ppm), multiplicity, coupling constant (Hz), and integration. ^13^C NMR data was reported in terms of chemical shift (δ ppm), multiplicity, and coupling constant (Hz). Mass (HRMS) analysis was obtained using Agilent 6200 Accurate-Mass TOF LC/MS (Agilent Technologies Co., Ltd., Santa Clara, California, USA). system with Electrospray Ionization (ESI). Aliphatic aldehydes **1** and (hetero)aryl iodides (**2a–2g** and **2j–2n**) were purchased from Energy-chemical (Shanghai, China), Atomax-chemical (Shenzhen, China), BLDpharm (Shanghai, China), Chemieliva (Chongqing, China), Enamine (Shanghai, China), Adamas-beta^®^ (Shanghai, China), TCI (Shanghai, China), J&K^@^ (Shanghai, China) or Sigma-Aldrich (Shanghai, China). Aryl iodide **2h** and **2i** were prepared by using 4-iodobenzoic acid with menthol and fenchol according to literature procedures [38].

### 3.2. Optimization of the Reaction Conditions

A 10 mL Schlenk tube was charged with methyl 4-iodobenzoate (**2a**, 104.81 mg, 0.4 mmol), L-phenylglycine (**TDG1**), 3-nitro-5-(trifluoromethyl)pyridin-2-ol (**L1**), Pd source (0.02 mmol), and AgTFA (0.3 mmol). The tube was evacuated and filled with N_2_ three times. Next, pivalaldehyde (**1a**, 22.0 μL, 0.2 mmol) and the solvents were added into the tube quickly. The reaction was then stirred vigorously at room temperature for 20 min before being heated to 100 °C for 24 h. After cooling to room temperature, the reaction mixture was diluted with EtOAc (15 mL), filtered through a pad of celite, and the filtrate was then concentrated in vacuo; the crude product was analyzed by ^1^H NMR in CDCl_3_. Yields are based on **1a**, determined by crude ^1^H NMR using dibromomethane as the internal standard. The residue was purified by flash chromatography on silica gel using petroleum ether/EtOAc (*v*/*v* = 10:1) as the eluent to yield products **3a-mono** and **3a-di**.

### 3.3. The Investigations of Transient Directing Groups

A 10 mL Schlenk tube was charged with methyl 4-iodobenzoate (**2a**, 104.81 mg, 0.4 mmol), transient directing groups (**TDG**, 0.08 mmol), 3-nitro-5-(trifluoromethyl)pyridin-2-ol (**L1**, 25.0 mg, 0.12 mmol), Pd(OAc)_2_ (4.53 mg, 0.02 mmol), and AgTFA (66.27 mg, 0.3 mmol). The tube was evacuated and filled with N_2_ three times. Next, pivalaldehyde (**1a**, 22.0 μL, 0.2 mmol) and the mixture of HFIP (1.5 mL) and HOAc (0.5 mL) were added into the tube quickly. The reaction was then stirred vigorously at room temperature for 20 min before being heated to 100 °C for 24 h. After cooling to room temperature, the reaction mixture was diluted with EtOAc (15 mL), filtered through a pad of celite, and the filtrate was then concentrated in vacuo; the crude product was analyzed by ^1^H NMR in CDCl_3_. Yields (**3a-mono** and **3a-di**) are based on **1a**, determined by crude ^1^H NMR using dibromomethane as the internal standard. 

### 3.4. The Investigations of 2-Pyridone Ligands

A 10 mL Schlenk tube was charged with methyl 4-iodobenzoate (**2a**, 104.81 mg, 0.4 mmol), L-phenylglycine (**TDG1**, 12.09 mg, 0.08 mmol), 2-pyridone ligands (**L1**, 0.12 mmol), Pd(OAc)_2_ (4.53 mg, 0.02 mmol), and AgTFA (66.27 mg, 0.3 mmol). The tube was evacuated and filled with N_2_ three times. Next, pivalaldehyde (**1a**, 22.0 μL, 0.2 mmol) and the mixture of HFIP (1.5 mL) and HOAc (0.5 mL) were added into the tube quickly. The reaction was then stirred vigorously at room temperature for 20 min before being heated to 100 °C for 24 h. After cooling to room temperature, the reaction mixture was diluted with EtOAc (15 mL), filtered through a pad of celite, and the filtrate was then concentrated in vacuo; the crude product was analyzed by ^1^H NMR in CDCl_3_. Yields (**3a-mono** and **3a-di**) are based on **1a**, determined by crude ^1^H NMR using dibromomethane as the internal standard.

### 3.5. Synthetic Procedures for the Synthesis of Compound **3**

A 10 mL Schlenk tube was charged with iodobenzene (**2**, 0.4 mmol), transient directing groups (**TDG1** or **TDG2**, 0.08 mmol), 2-pyridone ligands (**L1** or **L2**, 0.12 mmol), Pd(OAc)_2_ (4.53 mg, 0.02 mmol), and AgTFA (66.27 mg, 0.3 mmol). The tube was evacuated and filled with N_2_ three times. Next, aldehyde (**1**, 0.2 mmol) and the mixture of HFIP (1.5 mL) and HOAc (0.5 mL) were added into the tube quickly. The reaction was then stirred vigorously at room temperature for 20 min before being heated to 100 °C for 24 or 72 h. After cooling to room temperature, the reaction mixture was diluted with EtOAc (15 mL), filtered through a pad of celite, and the filtrate was then concentrated in vacuo; the residue was purified by flash chromatography on silica gel using petroleum ether/EtOAc as the eluent to yield the product **3**.

*Methyl 4-(2,2-dimethyl-3-oxopropyl)benzoate* (**3a-mono**). The title compound was prepared by using **TDG1** and **L1**, then isolated by flash chromatography on the silica gel by using mixed petroleum ether and ethyl acetate (*v*/*v* = 10:1). Colorless oil, 22.5 mg, yield: 51% (known compound [21]). ^1^H NMR (300 MHz, CDCl_3_) δ 9.50 (s, 1H), 7.88 (d, *J* = 8.3 Hz, 2H), 7.10 (d, *J* = 8.2 Hz, 2H), 3.83 (s, 3H), 2.76 (s, 2H), 0.98 (s, 6H). ^13^C NMR (75 MHz, CDCl_3_) δ 205.41, 167.00, 142.50, 130.33, 129.46, 128.53, 52.12, 46.95, 42.90, 21.45.

*Dimethyl 4,4′-(2-formyl-2-methylpropane-1,3-diyl)dibenzoate* (**3a-di**). The title compound was prepared by using **TDG1** and **L1**, then isolated by flash chromatography on the silica gel by using mixed petroleum ether and ethyl acetate (*v*/*v* = 10:1). White solid, 22.7 mg, yield: 32%. ^1^H NMR (300 MHz, CDCl_3_) δ 9.58 (s, 1H), 7.88 (d, *J* = 8.3 Hz, 4H), 7.09 (d, *J* = 8.3 Hz, 4H), 3.84 (s, 6H), 3.00 (d, *J* = 13.5 Hz, 2H), 2.69 (d, *J* = 13.5 Hz, 2H), 0.93 (s, 3H). ^13^C NMR (75 MHz, CDCl_3_) δ 200.17, 161.91, 136.75, 125.44, 124.60, 123.79, 47.18, 46.15, 37.59, 13.36. HRMS (ESI, *m*/*z*): calcd. For C_21_H_22_NaO_5_ [M + Na]^+^: 377.1358, found: 377.1358.

*Ethyl 4-(2,2-dimethyl-3-oxopropyl)benzoate* (**3b-mono**). The title compound was prepared by using **TDG1** and **L1**, then isolated by flash chromatography on the silica gel by using mixed petroleum ether and ethyl acetate (*v*/*v* = 10:1). Colorless oil, 23.4 mg, yield: 50% (known compound [21]). ^1^H NMR (400 MHz, CDCl_3_) δ 9.50 (s, 1H), 7.93–7.84 (m, 2H), 7.10 (d, *J* = 8.3 Hz, 2H), 4.32–4.26 (m, 2H), 2.76 (s, 2H), 1.31 (t, *J* = 7.1 Hz, 3H), 0.98 (s, 6H). ^13^C NMR (75 MHz, CDCl_3_) δ 204.34, 165.47, 141.30, 129.21, 128.36, 127.84, 59.87, 45.87, 41.87, 20.39, 13.29.

*Diethyl 4,4′-(2-formyl-2-methylpropane-1,3-diyl)dibenzoate* (**3b-di**). The title compound was prepared by using **TDG1** and **L1**, then isolated by flash chromatography on the silica gel by using mixed petroleum ether and ethyl acetate (*v*/*v* = 10:1). White solid, 24.5 mg, yield: 32%. ^1^H NMR (400 MHz, CDCl_3_) δ 9.58 (s, 1H), 7.88 (d, *J* = 8.3 Hz, 4H), 7.08 (d, *J* = 8.3 Hz, 4H), 4.32–4.26 (m, 4H), 2.99 (d, *J* = 13.5 Hz, 2H), 2.69 (d, *J* = 13.5 Hz, 2H), 1.31 (t, *J* = 7.1 Hz, 6H), 0.92 (s, 3H). ^13^C NMR (75 MHz, CDCl_3_) δ 205.16, 166.41, 141.62, 130.37, 129.55, 129.14, 60.98, 51.13, 42.58, 18.35, 14.35. HRMS (ESI, *m*/*z*): calcd. For C_23_H_27_O_5_ [M + H]^+^: 383.1853, found: 383.1838.

*4-(2,2-Dimethyl-3-oxopropyl)benzonitrile* (**3c-mono**). The title compound was prepared by using **TDG1** and **L1**, then isolated by flash chromatography on the silica gel by using mixed petroleum ether and ethyl acetate (*v*/*v* = 50:1). Colorless oil, 22.8 mg, yield: 61% (known compound [39]). ^1^H NMR (300 MHz, CDCl_3_) δ 9.56 (s, 1H), 7.57 (d, *J* = 8.0 Hz, 2H), 7.23 (d, *J* = 8.0 Hz, 2H), 2.85 (s, 2H), 1.06 (s, 6H). ^13^C NMR (75 MHz, CDCl_3_) δ 204.84, 142.86, 131.93, 131.06, 118.79, 110.57, 46.92, 42.82, 21.52.

*4,4′-(2-Formyl-2-methylpropane-1,3-diyl)dibenzonitrile* (**3c-di**). The title compound was prepared by using **TDG1** and **L1**, then isolated by flash chromatography on the silica gel by using mixed petroleum ether and ethyl acetate (*v*/*v* = 50:1). Colorless oil, 8.1 mg, yield: 14%. ^1^H NMR (300 MHz, CDCl_3_) δ 9.60 (s, 1H), 7.58 (d, *J* = 8.0 Hz, 4H), 7.20 (d, *J* = 8.0 Hz, 4H), 3.07 (d, *J* = 13.5 Hz, 2H), 2.75 (d, *J* = 13.5 Hz, 2H), 1.02 (s, 3H). ^13^C NMR (75 MHz, CDCl_3_) δ 204.25, 141.68, 132.14, 131.14, 118.58, 111.07, 51.09, 42.60, 18.47. HRMS (ESI, *m*/*z*): calcd. For C_19_H_16_N_2_NaO [M + Na]^+^: 311.1155, found: 311.1151.

*2,2-Dimethyl-3-(4-nitrophenyl)propanal* (**3d-mono**). The title compound was prepared by using **TDG1** and **L1**, then isolated by flash chromatography on the silica gel by using mixed petroleum ether and ethyl acetate (*v*/*v* = 50:1). Colorless oil, 31.1 mg, yield: 75% (known compound [21]). ^1^H NMR (300 MHz, CDCl_3_) δ 9.57 (s, 1H), 8.14 (d, *J* = 8.7 Hz, 2H), 7.29 (d, *J* = 8.7 Hz, 2H), 2.91 (s, 2H), 1.09 (s, 6H). ^13^C NMR (75 MHz, CDCl_3_) δ 204.78, 146.84, 145.06, 131.15, 123.39, 46.99, 42.46, 21.57.

*2-Methyl-2-(4-nitrobenzyl)-3-(4-nitrophenyl)propanal* (**3d-di**). The title compound was prepared by using **TDG1** and **L1**, then isolated by flash chromatography on the silica gel by using mixed petroleum ether and ethyl acetate (*v*/*v* = 50:1). Colorless oil, 6.6 mg, yield: 10% (known compound [18]). ^1^H NMR (300 MHz, CDCl_3_) δ 9.56 (s, 1H), 8.09 (d, *J* = 8.4 Hz, 4H), 7.20 (d, *J* = 5.6 Hz, 4H), 3.07 (d, *J* = 13.4 Hz, 2H), 2.75 (d, *J* = 13.5 Hz, 2H), 0.99 (s, 3H). ^13^C NMR (75 MHz, CDCl_3_) δ 204.10, 147.09, 143.74, 131.26, 123.61, 51.15, 42.30, 18.54.

*2,2-Dimethyl-3-(4-(trifluoromethyl)phenyl)propanal* (**3e-mono**). The title compound was prepared by using **TDG1** and **L1**, then isolated by flash chromatography on the silica gel by using mixed petroleum ether and ethyl acetate (*v*/*v* = 50:1). Colorless oil, 22.6 mg, yield: 49% (known compound [40]). ^1^H NMR (300 MHz, CDCl_3_) δ 9.50 (s, 1H), 7.45 (d, *J* = 7.9 Hz, 2H), 7.14 (d, *J* = 7.9 Hz, 2H), 2.77 (s, 2H), 0.99 (s, 6H). ^13^C NMR (101 MHz, CDCl_3_) δ 205.28, 141.24, 130.59, 128.90 (q, *J* = 32.5 Hz), 125.09 (q, *J* = 3.8 Hz), 124.24 (q, *J* = 272.0 Hz), 46.91, 42.64, 21.44. 

*2-Methyl-2-(4-(trifluoromethyl)benzyl)-3-(4-(trifluoromethyl)phenyl)propanal* (**3e-di**). The title compound was prepared by using **TDG1** and **L1**, then isolated by flash chromatography on the silica gel by using mixed petroleum ether and ethyl acetate (*v*/*v* = 50:1). Colorless oil, 16.5 mg, yield: 22% (known compound [40]). ^1^H NMR (300 MHz, CDCl_3_) δ 9.57 (s, 1H), 7.46 (d, *J* = 8.0 Hz, 4H), 7.13 (d, *J* = 8.0 Hz, 4H), 3.00 (d, *J* = 13.6 Hz, 2H), 2.69 (d, *J* = 13.6 Hz, 2H), 0.94 (s, 3H). ^13^C NMR (101 MHz, CDCl_3_) δ 204.97, 140.39, 130.70, 129.22 (q, *J* = 32.6 Hz), 125.28 (q, *J* = 3.7 Hz), 124.14 (q, *J* = 271.9 Hz), 51.05, 42.38, 18.32. 

*Methyl 3-(2,2-dimethyl-3-oxopropyl)benzoate* (**3f-mono**). The title compound was prepared by using **TDG1** and **L1**, then isolated by flash chromatography on the silica gel by using mixed petroleum ether and ethyl acetate (*v*/*v* = 10:1). Colorless oil, 20.3 mg, yield: 46% (known compound [21]). ^1^H NMR (300 MHz, CDCl_3_) δ 9.52 (s, 1H), 7.84 (d, *J* = 7.4 Hz, 1H), 7.72 (s, 1H), 7.31–7.21 (m, 2H), 3.84 (s, 3H), 2.77 (s, 2H), 0.99 (s, 6H). ^13^C NMR (75 MHz, CDCl_3_) δ 205.49, 167.06, 137.35, 134.79, 131.26, 130.10, 128.27, 127.88, 52.18, 46.90, 42.71, 21.39. 

*Dimethyl 3,3′-(2-formyl-2-methylpropane-1,3-diyl)dibenzoate* (**3f-di**). The title compound was prepared by using **TDG1** and **L1**, then isolated by flash chromatography on the silica gel by using mixed petroleum ether and ethyl acetate (*v*/*v* = 10:1). Colorless oil, 22.7 mg, yield: 32%. ^1^H NMR (300 MHz, CDCl_3_) δ 9.61 (s, 1H), 7.84 (d, *J* = 7.6 Hz, 2H), 7.70 (s, 2H), 7.32–7.19 (m, 4H), 3.84 (s, 6H), 2.99 (d, *J* = 13.6 Hz, 2H), 2.70 (d, *J* = 13.6 Hz, 2H), 0.94 (s, 3H). ^13^C NMR (75 MHz, CDCl_3_) δ 205.23, 166.94, 136.73, 134.85, 131.38, 130.24, 128.41, 128.09, 52.17, 51.07, 42.31, 18.25. HRMS (ESI, *m*/*z*): calcd. For C_21_H_22_NaO_5_ [M + Na]^+^: 377.1354, found: 377.1358.

*2,2-Dimethyl-3-(3-nitrophenyl)propanal* (**3g-mono**). The title compound was prepared by using **TDG1** and **L1**, then isolated by flash chromatography on the silica gel by using mixed petroleum ether and ethyl acetate (*v*/*v* = 50:1). Colorless oil, 22.0 mg, yield: 53%. ^1^H NMR (300 MHz, CDCl_3_) δ 9.50 (s, 1H), 8.02–8.00 (m, 1H), 7.92 (s, 1H), 7.38 (d, *J* = 5.0 Hz, 2H), 2.84 (s, 2H), 1.01 (s, 6H). ^13^C NMR (75 MHz, CDCl_3_) δ 204.78, 148.11, 139.21, 136.47, 129.10, 124.98, 121.76, 46.88, 42.26, 21.47. HRMS (ESI, *m*/*z*): calcd. For C_11_H_14_NO_3_ [M + H]^+^: 208.0968, found: 208.0968.

*2-Methyl-2-(3-nitrobenzyl)-3-(3-nitrophenyl)propanal* (**3g-di**). The title compound was prepared by using **TDG1** and **L1**, then isolated by flash chromatography on the silica gel by using mixed petroleum ether and ethyl acetate (*v*/*v* = 50:1). Yellow solid, 11.8 mg, yield: 18%. ^1^H NMR (300 MHz, CDCl_3_) δ 9.59 (s, 1H), 8.06 (d, *J* = 7.6 Hz, 2H), 7.91 (s, 2H), 7.47–7.31 (m, 4H), 3.07 (d, *J* = 13.7 Hz, 2H), 2.77 (d, *J* = 13.7 Hz, 2H), 1.02 (s, 3H). ^13^C NMR (75 MHz, CDCl_3_) δ 204.05, 148.21, 138.07, 136.47, 129.40, 125.11, 122.18, 51.04, 41.98, 18.47. HRMS (ESI, *m*/*z*): calcd. For C_17_H_16_N_2_NaO_5_ [M + Na]^+^: 351.0951, found: 351.0933.

*2,2-Dimethyl-3-(6-(trifluoromethyl)yridine-3-yl)propanal* (**3h**). The title compound was prepared by using **TDG1** and **L1**, then isolated by flash chromatography on the silica gel by using mixed petroleum ether and ethyl acetate (*v*/*v* = 10:1). Yellow oil, 23.1 mg, yield: 50% (known compound [40]). ^1^H NMR (400 MHz, CDCl_3_) δ 9.47 (s, 1H), 8.44 (s, 1H), 7.60–7.53 (m, 2H), 2.81 (s, 2H), 1.02 (s, 6H). ^13^C NMR (101 MHz, CDCl_3_) δ 204.47, 151.43, 146.49 (q, *J* = 34.7 Hz), 139.02, 136.33, 121.60 (q, *J* = 273.8 Hz), 119.96 (q, *J* = 2.7 Hz), 46.81, 39.27, 21.47. 

*(1S,2S,5S)-2-Isopropyl-5-methylcyclohexyl 4-(2,2-dimethyl-3-oxopropyl)benzoate* (**3i-mono**). The title compound was prepared by using **TDG1** and **L1**, then isolated by flash chromatography on the silica gel by using mixed petroleum ether and ethyl acetate (*v*/*v* = 10:1). Colorless oil, 26.2 mg, yield: 38%. ^1^H NMR (300 MHz, CDCl_3_) δ 9.51 (s, 1H), 7.88 (d, *J* = 8.1 Hz, 2H), 7.10 (d, *J* = 8.1 Hz, 2H), 4.89–4.80 (m, 1H), 2.76 (s, 2H), 2.06–2.02 (m, 1H), 1.91–1.86 (m, 1H), 1.67–1.64 (m, 2H), 1.54–1.43 (m, 2H), 1.21–0.96 (m, 9H), 0.86–0.83 (m, 6H), 0.72 (d, *J* = 6.9 Hz, 3H). ^13^C NMR (75 MHz, CDCl_3_) δ 205.38, 165.98, 142.22, 130.24, 129.43, 129.27, 74.79, 47.29, 46.93, 42.95, 40.99, 34.34, 31.45, 26.47, 23.62, 22.05, 21.48, 21.44, 20.80, 16.50. HRMS (ESI, *m*/*z*): calcd. For C_22_H_32_NaO_3_ [M + Na]^+^: 367.2244, found: 367.2249.

*(1S,2S,5S)-2-isopropyl-5-methylcyclohexyl 4-(2-formyl-3-(4-((((1R,2R,5R)-2-isopropyl-5-methylcyclohexyl)oxy)carbonyl)phenyl)-2-methylpropyl)benzoate* (**3i-di**). The title compound was prepared by using **TDG1** and **L1**, then isolated by flash chromatography on the silica gel by using mixed petroleum ether and ethyl acetate (*v*/*v* = 10:1). Colorless oil, 29.0 mg, yield: 24%. ^1^H NMR (300 MHz, CDCl_3_) δ 9.59 (s, 1H), 7.88 (d, *J* = 8.0 Hz, 4H), 7.08 (d, *J* = 8.0 Hz, 4H), 4.89–4.80 (m, 2H), 3.00 (dd, *J* = 13.5, 2.0 Hz, 2H), 2.70 (dd, *J* = 13.5, 2.6 Hz, 2H), 2.06–2.02 (m, 2H), 1.90–1.86 (m, 2H), 1.67–1.64 (m, 4H), 1.53–1.43 (m, 4H), 1.21–0.99 (m, 6H), 0.94 (s, 3H), 0.96–0.83 (m, 12H), 0.72 (d, *J* = 6.9 Hz, 6H). ^13^C NMR (101 MHz, CDCl_3_) δ 205.25, 165.91, 141.51, 130.36, 129.57, 129.49, 74.86, 51.20, 47.27, 42.66, 40.98, 34.32, 31.46, 26.45, 23.57, 22.08, 20.83, 18.34, 16.49. HRMS (ESI, *m*/*z*): calcd. For C_39_H_55_O_5_ [M+H]^+^: 603.4044, found: 603.4031.

*(1R,4S)-1,3,3-Trimethylbicyclo[2.2.1]heptan-2-yl 4-(2,2-dimethyl-3-oxopropyl)benzoate* (**3j-mono**). The title compound was prepared by using **TDG1** and **L1**, then isolated by flash chromatography on the silica gel by using mixed petroleum ether and ethyl acetate (*v*/*v* = 10:1). Colorless oil, 34.2 mg, yield: 50% (known compound [21]). ^1^H NMR (300 MHz, CDCl_3_) δ 9.51 (s, 1H), 7.90 (d, *J* = 8.0 Hz, 2H), 7.11 (d, *J* = 8.0 Hz, 2H), 4.53 (s, 1H), 2.77 (s, 2H), 1.90–1.81 (m, 1H), 1.74–1.67 (m, 2H), 1.61–1.54 (m, 1H), 1.50–1.39 (m, 1H), 1.17 (d, *J* = 8.2 Hz, 2H), 1.11 (s, 3H), 1.03 (s, 3H), 1.00 (s, 6H), 0.77 (s, 3H). ^13^C NMR (75 MHz, CDCl_3_) δ 205.38, 166.75, 142.32, 130.32, 129.41, 129.10, 86.65, 48.64, 48.43, 46.95, 42.96, 41.47, 39.85, 29.76, 26.89, 25.92, 21.47, 20.33, 19.50.

*(1S,4R)-1,3,3-Trimethylbicyclo[2.2.1]heptan-2-yl 4-(2-formyl-2-methyl-3-(4-((((1R,4S)-1,3,3-trimethylbicyclo[2.2.1]heptan-2-yl)oxy)carbonyl)phenyl)propyl)benzoate* (**3j-di**). The title compound was prepared by using **TDG1** and **L1**, then isolated by flash chromatography on the silica gel by using mixed petroleum ether and ethyl acetate (*v*/*v* = 10:1). White solid, 36.0 mg, yield: 30%. ^1^H NMR (300 MHz, CDCl_3_) δ 9.60 (s, 1H), 7.90 (d, *J* = 8.0 Hz, 4H), 7.10 (d, *J* = 8.0 Hz, 4H), 4.53 (s, 2H), 3.01 (d, *J* = 13.5 Hz, 2H), 2.71 (d, *J* = 13.5 Hz, 2H), 1.89–1.80 (m, 2H), 1.74–1.67 (m, 4H), 1.60–1.58 (m, 2H), 1.50–1.40 (m, 2H), 1.17 (d, *J* = 10.0 Hz, 4H), 1.10 (s, 6H), 1.03 (s, 6H), 0.95 (s, 3H), 0.77 (s, 6H). ^13^C NMR (75 MHz, CDCl_3_) δ 205.18, 166.65, 141.61, 130.43, 129.54, 129.34, 86.71, 51.20, 48.64, 48.43, 42.65, 41.47, 39.85, 29.76, 26.89, 25.92, 20.35, 19.51, 18.38. HRMS (ESI, *m*/*z*): calcd. For C_39_H_50_NaO_5_ [M + Na]^+^: 621.3550, found: 621.3543.

Compound **3k** is a mixture of *2,2-dimethyl-3-phenylpropanal* (**3k-mono**), 2-benzyl-2-methyl-3-phenylpropanal (**3k-di**), and 2,2-dibenzyl-3-phenylpropanal (**3k-tri**). **3k-mono:3k-di:3k-tri** = 1.0:0.76:0.35. The compound **3k** was prepared by using **TDG2** and **L2**, then isolated by flash chromatography on the silica gel by using mixed petroleum ether and ethyl acetate (*v*/*v* = 100:1). Colorless oil, 28.5 mg, yield: 66% (known compound [18]). *2,2-Dimethyl-3-phenylpropanal* (**3k-mono**): ^1^H NMR (300 MHz, CDCl_3_) δ 9.51 (s, 1H), 7.22–7.00 (m, 5H), 2.70 (s, 2H), 0.98 (s, 6H). *2-Benzyl-2-methyl-3-phenylpropanal* (**3k-di**): ^1^H NMR (300 MHz, CDCl_3_) δ 9.61 (s, 1H), 7.22–7.00 (m, 5H), 2.95 (d, *J* = 13.6 Hz, 2H), 2.64 (d, *J* = 13.6 Hz, 2H), 0.91 (s, 3H). *2,2-Dibenzyl-3-phenylpropanal* (**3k-tri**): ^1^H NMR (300 MHz, CDCl_3_) δ 9.70 (s, 1H), 7.22–7.00 (m, 5H), 2.85 (s, 6H). ^13^C NMR (75 MHz, CDCl_3_) δ 206.86, 206.22, 206.01, 136.92, 136.62, 136.59, 130.59, 130.37, 130.26, 128.30, 128.26, 128.17, 126.66, 126.55, 53.56, 51.24, 46.96, 43.23, 42.84, 40.22, 21.40, 18.17.

*3-(4-Methoxyphenyl)-2,2-dimethylpropanal* (**3l-mono**). The title compound was prepared by using **TDG2** and **L2**, then isolated by flash chromatography on the silica gel by using mixed petroleum ether and ethyl acetate (*v*/*v* = 50:1). Colorless oil, 11.5 mg, yield: 30% (known compound [18]). ^1^H NMR (300 MHz, CDCl_3_) δ 9.50 (s, 1H), 6.93 (d, *J* = 8.1 Hz, 2H), 6.73 (d, *J* = 8.1 Hz, 2H), 3.71 (s, 3H), 2.65 (s, 2H), 0.96 (s, 6H). ^13^C NMR (75 MHz, CDCl_3_) δ 206.24, 158.32, 131.18, 128.90, 113.58, 55.22, 47.04, 42.38, 21.33.

*2-(4-Methoxybenzyl)-3-(4-methoxyphenyl)-2-methylpropana*l (**3l-di**). The title compound was prepared by using **TDG2** and **L2**, then isolated by flash chromatography on the silica gel by using mixed petroleum ether and ethyl acetate (*v*/*v* = 50:1). Colorless oil, 13.1 mg, yield: 22% (known compound [18]). ^1^H NMR (300 MHz, CDCl_3_) δ 9.58 (s, 1H), 6.92 (d, *J* = 8.1 Hz, 4H), 6.72 (d, *J* = 8.3 Hz, 4H), 3.70 (s, 6H), 2.88 (d, *J* = 13.8 Hz, 2H), 2.56 (d, *J* = 13.8 Hz, 2H), 0.88 (s, 3H). ^13^C NMR (75 MHz, CDCl_3_) δ 206.66, 158.35, 131.30, 128.62, 113.64, 55.22, 51.46, 41.91, 18.06.

*2,2-Bis(4-methoxybenzyl)-3-(4-methoxyphenyl)propanal* (**3l-tri**). The title compound was prepared by using **TDG2** and **L2**, then isolated by flash chromatography on the silica gel by using mixed petroleum ether and ethyl acetate (*v*/*v* = 50:1). Colorless oil, 8.1 mg, yield: 10% (known compound [18]). ^1^H NMR (300 MHz, CDCl_3_) δ 9.74 (s, 1H), 7.02 (d, *J* = 8.3 Hz, 6H), 6.80 (d, *J* = 8.4 Hz, 6H), 3.78 (s, 9H), 2.84 (s, 6H). ^13^C NMR (75 MHz, CDCl_3_) δ 207.40, 158.28, 131.53, 128.62, 113.64, 55.22, 53.81, 39.23.

Compound **3m** is a mixture of *2,2-dimethyl-3-(p-tolyl)propanal* (**3m-mono**), *2-methyl*-*2-(4-methylbenzyl)-3-(p-tolyl)propanal* (**3m-di**), and *2,2-bis(4-methylbenzyl)-3-(p-tolyl)propanal* (**3m-tri**). **3m-mono:3m-di:3m-tri** = 1.0:0.71:0.42. The compound **3m** was prepared by using **TDG2** and **L2**, then isolated by flash chromatography on the silica gel by using mixed petroleum ether and ethyl acetate (*v*/*v* = 100:1). Colorless oil, 33.0 mg, yield:68% (known compound [21]). *2,2-Dimethyl-3-(p-tolyl)propanal* (**3m-mono**): ^1^H NMR (300 MHz, CDCl_3_) δ 9.51 (s, 1H), 7.18–6.88 (m, 4H), 2.66 (s, 2H), 2.23 (s, 3H), 0.97 (s, 6H). *2-Methyl-2-(4-methylbenzyl)-3-(p-tolyl)propanal* (**3m-di**): ^1^H NMR (300 MHz, CDCl_3_) δ 9.60 (s, 1H), 7.18–6.88 (m, 4H), 2.90 (d, *J* = 13.6 Hz, 2H), 2.58 (d, *J* = 13.6 Hz, 2H), 2.23 (s, 6H), 0.88 (s, 3H). *2,2-bis(4-methylbenzyl)-3-(p-tolyl)propanal* (**3m-tri**): ^1^H NMR (300 MHz, CDCl_3_) δ 9.67 (s, 1H), 7.18–6.88 (m, 4H), 2.79 (s, 6H), 2.23 (s, 9H). ^13^C NMR (75 MHz, CDCl_3_) δ 207.30, 206.50, 206.16, 136.16, 136.08, 133.75, 133.58, 133.51, 130.48, 130.22, 130.12, 128.93, 128.86, 53.59, 51.30, 46.95, 42.87, 42.43, 39.71, 21.37, 21.01, 18.10.

*3-(4-Bromophenyl)-2,2-dimethylpropanal* (**3n-mono**). The title compound was prepared by using **TDG2** and **L2**, then isolated by flash chromatography on the silica gel by using mixed petroleum ether and ethyl acetate (*v*/*v* = 100:1). Colorless oil, 24.0 mg, yield: 50% (known compound [18]). ^1^H NMR (400 MHz, CDCl_3_) δ 9.48 (s, 1H), 7.32 (d, *J* = 8.4 Hz, 2H), 6.90 (d, *J* = 8.3 Hz, 2H), 2.66 (s, 2H), 0.97 (s, 6H). ^13^C NMR (75 MHz, CDCl_3_) δ 205.52, 135.97, 131.95, 131.27, 120.57, 46.80, 42.39, 21.39.

*2-(4-Bromobenzyl)-3-(4-bromophenyl)-2-methylpropanal* (**3n-di**). The title compound was prepared by using **TDG2** and **L2**, then isolated by flash chromatography on the silica gel by using mixed petroleum ether and ethyl acetate (*v*/*v* = 100:1). Colorless oil, 19.8 mg, yield: 25% (known compound [18]). ^1^H NMR (400 MHz, CDCl_3_) δ 9.54 (s, 1H), 7.32 (d, *J* = 8.3 Hz, 4H), 6.87 (d, *J* = 8.4 Hz, 4H), 2.87 (d, *J* = 13.7 Hz, 2H), 2.56 (d, *J* = 13.7 Hz, 2H), 0.89 (s, 3H). ^13^C NMR (75 MHz, CDCl_3_) δ 205.44, 135.33, 132.03, 131.42, 120.84, 50.91, 41.99, 18.23.

Compound **3o** is a mixture of *3-(2-fluorophenyl)-2,2-dimethylpropanal* (**3o-mono**), 2-(2-fluorobenzyl)-3-(2-fluorophenyl)-2-methylpropanal (**3o-di**), and 2,2-bis(2-fluorobenzyl)-3-(2-fluorophenyl)propanal (**3o-tri**). The compound **3o** was prepared by using **TDG2** and **L2**, then isolated by flash chromatography on the silica gel by using mixed petroleum ether and ethyl acetate (*v*/*v* = 100:1). **3o-mono:3o-di:3o-tri** = 1.0:0.77:0.28. Colorless oil, 29.0 mg, yield: 60% (known compound [18]). *3-(2-Fluorophenyl)-2,2-dimethylpropanal* (**3o-mono**): ^1^H NMR (400 MHz, CDCl_3_) δ 9.51 (d, *J* = 1.3 Hz, 1H), 7.17–6.92 (m, 4H), 2.76 (d, *J* = 1.1 Hz, 2H), 1.00 (s, 6H). *2-(2-Fluorobenzyl)-3-(2-fluorophenyl)-2-methylpropanal* (**3o-di**): ^1^H NMR (400 MHz, CDCl_3_) δ 9.61 (t, *J* = 1.9 Hz, 1H), 7.17–6.92 (m, 4H), 3.00 (d, *J* = 13.7 Hz, 2H), 2.76 (d, *J* = 13.7 Hz, 2H), 0.91 (s, 3H). *2,2-Bis(2-fluorobenzyl)-3-(2-fluorophenyl)propanal* (**3o-tri**): ^1^H NMR (400 MHz, CDCl_3_) δ 9.57 (d, *J* = 1.7 Hz, 1H), 7.17–6.92 (m, 4H), 2.91 (s, 6H). ^13^C NMR (75 MHz, CDCl_3_) δ 204.44, 203.86, 203.16, 161.99, 161.87, 161.83, 158.75, 158.63, 158.58, 131.92, 131.86, 131.63, 131.58, 131.52, 127.76, 127.65, 127.55, 127.44, 122.99, 122.92, 122.87, 122.80, 122.75, 122.65, 122.44, 114.62, 114.54, 114.31, 114.23, 52.75, 50.58, 46.09, 34.71, 34.42, 31.66, 20.23, 16.27.

*Methyl 4-(2-formyl-2-propylpentyl)benzoate* (**3p**). The title compound was prepared by using **TDG1** and **L1**, then isolated by flash chromatography on the silica gel by using mixed petroleum ether and ethyl acetate (*v*/*v* = 10:1). Colorless oil, 36.0 mg, yield: 65%. ^1^H NMR (400 MHz, CDCl_3_) δ 9.47 (s, 1H), 7.86 (d, *J* = 7.9 Hz, 2H), 7.07 (d, *J* = 7.9 Hz, 2H), 3.83 (s, 3H), 2.81 (s, 2H), 1.41–1.31 (m, 4H), 1.24–1.18 (m, 4H), 0.83 (t, *J* = 7.1 Hz, 6H). ^13^C NMR (75 MHz, CDCl_3_) δ 206.57, 166.99, 142.77, 130.06, 129.51, 128.43, 53.59, 52.09, 38.51, 34.32, 31.75, 25.77, 23.25, 16.98, 14.62, 13.97, 1.04. HRMS (ESI, *m*/*z*): calcd. for C_17_H_25_O_3_ [M + H]^+^: 277.1798, found: 277.1789.

*Methyl 4-(2-formyl-2-propylhexyl)benzoate* (**3q**). The title compound was prepared by using **TDG1** and **L1**, then isolated by flash chromatography on the silica gel by using mixed petroleum ether and ethyl acetate (*v*/*v* = 10:1). Colorless oil, 39.0 mg, yield: 67%. ^1^H NMR (300 MHz, CDCl_3_) δ 9.54 (s, 1H), 7.94 (d, *J* = 7.9 Hz, 2H), 7.14 (d, *J* = 7.9 Hz, 2H), 3.90 (s, 3H), 2.88 (s, 2H), 1.52–1.27 (m, 10H), 0.90 (t, *J* = 6.9 Hz, 6H). ^13^C NMR (75 MHz, CDCl_3_) δ 205.50, 165.92, 141.71, 128.99, 128.45, 127.36, 52.53, 51.02, 37.43, 33.24, 30.67, 24.70, 22.18, 15.91, 13.56, 12.90. HRMS (ESI, *m*/*z*): calcd. for C_18_H_27_O_3_ [M + H]^+^: 291.1955, found: 291.1948.

*Methyl 4-(2-formyl-2-propylheptyl)benzoate* (**3r**). The title compound was prepared by using **TDG1** and **L1**, then isolated by flash chromatography on the silica gel by using mixed petroleum ether and ethyl acetate (*v*/*v* = 10:1). Colorless oil, 37.7 mg, yield: 62%. ^1^H NMR (300 MHz, CDCl_3_) δ 9.47 (s, 1H), 7.86 (d, *J* = 8.2 Hz, 2H), 7.07 (d, *J* = 8.2 Hz, 2H), 3.83 (s, 3H), 2.81 (s, 2H), 1.45–1.18 (m, 12H), 0.85–0.79 (m, 6H). ^13^C NMR (75 MHz, CDCl_3_) δ 205.52, 165.96, 141.76, 129.02, 128.47, 127.39, 52.60, 51.04, 37.45, 33.30, 31.32, 31.02, 22.25, 21.44, 15.95, 13.58, 13.00. HRMS (ESI, *m*/*z*): calcd. for C_19_H_28_NaO_3_ [M + Na]^+^: 327.1931, found: 327.1921.

*Methyl 4-((1-formylcyclohexyl)methyl)benzoate* (**3s**). The title compound was prepared by using **TDG1** and **L1**, then isolated by flash chromatography on the silica gel by using mixed petroleum ether and ethyl acetate (*v*/*v* = 10:1). Colorless oil, 28.6 mg, yield: 55% (known compound [21]). ^1^H NMR (300 MHz, CDCl_3_) δ 9.45 (s, 1H), 7.86 (d, *J* = 8.3 Hz, 2H), 7.06 (d, *J* = 8.3 Hz, 2H), 3.83 (s, 3H), 2.70 (s, 2H), 1.85–1.81 (m, 2H), 1.59–1.46 (m, 3H), 1.29–1.17 (m, 5H). ^13^C NMR (75 MHz, CDCl_3_) δ 205.78, 165.94, 140.74, 129.25, 128.38, 127.52, 51.04, 49.67, 42.25, 30.14, 24.47, 21.58. 

*Methyl 4-(2-(4-chlorophenyl)-2-methyl-3-oxopropyl)benzoate* (**3t**). The title compound was prepared by using **TDG2** and **L1**, then isolated by flash chromatography on the silica gel by using mixed petroleum ether and ethyl acetate (*v*/*v* = 10:1). Colorless oil, 31.7 mg, yield: 50% (known compound [20]). ^1^H NMR (300 MHz, CDCl_3_) δ 9.49 (s, 1H), 7.74 (d, *J* = 7.7 Hz, 2H), 7.26 (d, *J* = 8.1 Hz, 2H), 6.98 (d, *J* = 8.1 Hz, 2H), 6.76 (d, *J* = 7.8 Hz, 2H), 3.80 (s, 3H), 3.12 (s, 2H), 1.30 (s, 3H). ^13^C NMR (75 MHz, CDCl3) δ 200.91, 166.98, 141.98, 137.10, 133.79, 130.43, 129.18, 129.03, 128.99, 128.51, 77.49, 77.06, 76.64, 54.65, 52.09, 42.70, 18.14.

*Methyl 4-(2-(2-bromophenyl)-2-methyl-3-oxopropyl)benzoate* (**3u**). The title compound was prepared by using **TDG2** and **L1**, then isolated by flash chromatography on the silica gel by using mixed petroleum ether and ethyl acetate (*v*/*v* = 10:1). Yellow oil, 44.0 mg, yield: 61% (known compound [20]). ^1^H NMR (300 MHz, CDCl_3_) δ 9.89 (s, 1H), 7.76 (d, *J* = 8.0 Hz, 2H), 7.70–7.67 (m, 1H), 7.23–7.15 (m, 2H), 6.82–6.79 (m, 1H), 6.73 (d, *J* = 8.0 Hz, 2H), 3.87 (s, 3H), 3.71 (d, *J* = 13.5 Hz, 1H), 3.29 (d, *J* = 13.5 Hz, 1H), 1.32 (s, 3H). ^13^C NMR (75 MHz, CDCl_3_) δ 202.20, 167.05, 142.29, 139.00, 134.43, 130.63, 130.52, 129.63, 128.95, 128.31, 127.57, 123.51, 55.95, 52.03, 39.13, 20.74.

*Methyl 4-(3-(benzyloxy)-2-formyl-2-methylpropyl)benzoate* (**3v-mono**). The title compound was prepared by using **TDG1** and **L1**, then isolated by flash chromatography on the silica gel by using mixed petroleum ether and ethyl acetate (*v*/*v* = 10:1). Colorless oil, 37.2 mg, yield: 57% (known compound [21]). ^1^H NMR (300 MHz, CDCl_3_) δ 9.58 (s, 1H), 7.84 (d, *J* = 8.2 Hz, 2H), 7.34–7.20 (m, 5H), 7.09 (d, *J* = 8.2 Hz, 2H), 4.42 (s, 2H), 3.82 (s, 3H), 3.29 (dd, *J* = 27.8, 9.3 Hz, 2H), 2.88 (dd, *J* = 52.1, 13.4 Hz, 2H), 0.90 (s, 3H). ^13^C NMR (75 MHz, CDCl3) δ 204.63, 167.01, 142.22, 137.72, 130.47, 129.46, 128.50, 127.88, 127.74, 73.38, 72.08, 52.10, 51.13, 37.70, 16.68. 

*Dimethyl 4,4′-(2-((benzyloxy)methyl)-2-formylpropane-1,3-diyl)dibenzoate* (**3v-di**). The title compound was prepared by using **TDG1** and **L1**, then isolated by flash chromatography on the silica gel by using mixed petroleum ether and ethyl acetate (*v*/*v* = 10:1). White solid, 16.6 mg, yield: 18%. ^1^H NMR (400 MHz, CDCl_3_) δ 9.67 (s, 1H), 7.91 (d, *J* = 8.2 Hz, 4H), 7.42–7.32 (m, 5H), 7.14 (d, *J* = 8.2 Hz, 4H), 4.45 (s, 2H), 3.90 (s, 6H), 3.29 (s, 2H), 3.11 (d, *J* = 13.6 Hz, 2H), 2.97 (d, *J* = 13.6 Hz, 2H). ^13^C NMR (101 MHz, CDCl_3_) δ 204.31, 166.90, 141.49, 137.51, 130.31, 129.65, 128.80, 128.56, 128.03, 73.49, 68.02, 55.57, 52.14, 38.53. HRMS (ESI, *m/z*): calcd. for C_28_H_29_O_6_ [M + H]^+^: 461.1959, found: 461.1949.

*Methyl 4-(3-((4-chlorobenzyl)oxy)-2-formyl-2-methylpropyl)benzoate* (**3w-mono**). The title compound was prepared by using **TDG2** and **L1**, then isolated by flash chromatography on the silica gel by using mixed petroleum ether and ethyl acetate (*v*/*v* = 10:1). White solid, 38.2 mg, yield: 53% (known compound [21]). ^1^H NMR (300 MHz, CDCl_3_) δ 9.58 (s, 1H), 7.86 (d, *J* = 8.1 Hz, 2H), 7.27 (d, *J* = 8.3 Hz, 2H), 7.18 (d, *J* = 8.3 Hz, 2H), 7.10 (d, *J* = 8.1 Hz, 2H), 4.38 (s, 2H), 3.84 (s, 3H), 3.29 (dd, *J* = 25.7, 9.3 Hz, 2H), 2.88 (dd, *J* = 49.0, 13.4 Hz, 2H), 0.92 (s, 3H). ^13^C NMR (75 MHz, CDCl_3_) δ 204.45, 166.99, 142.04, 136.20, 133.63, 130.40, 129.49, 128.95, 128.66, 128.60, 72.61, 72.28, 52.11, 51.11, 37.87, 16.76. 

*Dimethyl 4,4′-(2-(((4-chlorobenzyl)oxy)methyl)-2-formylpropane-1,3-diyl)dibenzoate* (**3w-di**). The title compound was prepared by using **TDG2** and **L1**, then isolated by flash chromatography on the silica gel by using mixed petroleum ether and ethyl acetate (*v*/*v* = 10:1). White solid, 12.0 mg, yield: 12%. ^1^H NMR (400 MHz, CDCl_3_) δ 9.68 (s, 1H), 7.92 (d, *J* = 8.2 Hz, 4H), 7.36 (d, *J* = 8.4 Hz, 2H), 7.27 (d, *J* = 7.1 Hz, 2H), 7.14 (d, *J* = 8.2 Hz, 4H), 4.40 (s, 2H), 3.91 (s, 6H), 3.28 (s, 2H), 3.12 (d, *J* = 13.6 Hz, 2H), 2.97 (d, *J* = 13.6 Hz, 2H). ^13^C NMR (101 MHz, CDCl_3_) δ 204.12, 166.88, 141.37, 135.98, 133.81, 130.25, 129.68, 129.23, 128.86, 128.73, 72.66, 68.21, 55.58, 52.19, 38.52. HRMS (ESI, *m/z*): calcd. for C_28_H_27_ClNaO_6_ [M + H]^+^: 495.1569, found: 495.1561.

## 4. Conclusions

In summary, we have developed a palladium-catalyzed direct methyl *β*-C(sp^3^)–H arylation reaction of tertiary aliphatic aldehydes with the commercially available *α*-amino acids as the transient directing groups and 2-pyridones as external ligands. Furthermore, a good functional group compatibility was observed in this catalytic cycle, and a variety of aryl iodides were efficiently coupled with different tertiary aliphatic aldehydes, providing the desired arylated aldehydes in moderate to good yields. A further application study is currently ongoing in our laboratory.

## Data Availability

Data are contained within the article and Appendix A.

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
