# Peer review of "Palladium-Catalyzed β-C(sp3)–H Bond Arylation of Tertiary Aldehydes Facilitated by 2-Pyridone Ligands"

_molecules, 2024, doi:10.3390/molecules29010259_

Round 1
Reviewer 1 Report
Comments and Suggestions for Authors
The manuscript by Ge and Yang describes palladium-catalyzed direct β-C(sp3) –H arylation of tertiary aliphatic aldehydes by using an α-amino acid as a transient directing group. The 2-pyridone ligand played a crucial role in the reaction. The reaction provides a novel method for the synthesis of arylated tertiary aliphatic aldehydes. I recommend this manuscript to be accepted for publications in molecules after the minor revisions as below.
1. Line 90, “optimaltransient” should be “optimal transient”.
2. Line 107, “delighted” should be “delight”.
3. The corresponding references should be cited for “previous reports” in line 140.
Comments on the Quality of English Languagesome typos can be found in the manuscript.
Author Response
- Reviewer comment: “Line 90, “optimaltransient” should be “optimal transient””
Response: Thanks to the reviewer for pointing this out. In the revised manuscript, the text “optimaltransient” has been corrected to “optimal transient”.
- Reviewer comment: “Line 107, “delighted” should be “delight””
Response: Thanks to the reviewer for pointing this out. In the revised manuscript, the text “delighted” has been corrected to “delight”.
- Reviewer comment: “The corresponding references should be cited for “previous reports” in line 140”
Response: Thanks to the reviewer for pointing this out. In the revised manuscript, the corresponding references have been cited for “previous reports”.
Reviewer 2 Report
Comments and Suggestions for Authors
This manuscript written by Yang and Ge et al. describes palladium-catalyzed C–H arylation of tertiary aldehyde at β-position. Pival aldehyde could be converted to the mixture of mono- and di-arylated product in the presence of palladium-hydroxypyridine catalyst and phenyl glycine or simple glycine as transient directing groups. Selective mono-arylation was possible for α-methyl-α,α-dialkyl or α-aryl-α,α-dimethyl acetaldehydes.
The reviewer thinks this catalytic system is good enough to be published in Molecules after responding the following comments for minor revision.
1) Reaction in HFIP/AcOH (1/1) showed better selectivity for mono-arylation product than the reaction in HFIP/AcOH (3/1) (Table 2, Entry 3). Is it possible to increase the mono-selectivity by optimizing the amount of TDG1 and L1 on the reaction in HFIP/AcOH (1/1)?
2) Tri-arylation products were obtained in the case of not electron-deficient iodoarene in Scheme 4. Could you provide the reasonable discussion on this trend?
3) The authors reported DFT-supported reaction mechanism for similar methylene β-C–H arylation of primary aliphatic aldehyde in Chem. Sci. 2022, 13, 5938. In this mechanism, hydroxypyridine ligand dissociates prior to oxidative addition of aryl iodide. In contrast, Scheme 6 depicts the oxidative addition of aryl iodide to hydroxypyridine-coordinated palladium(II) complex C. In general, oxidative addition of aryl iodide to palladium complex occurs thorough insertion of palladium to carbon-iodine bond. The previously proposed mechanism should also be applied to the reaction in this paper unless any significant evidence for Scheme 6 in the present form. Furthermore, abstraction of iodo ligand by silver(I) salt prior to reductive elimination to form C–C bond is proposed in the previous paper (Chem. Sci. 2022), which conflicts to the mechanism in Scheme 6.
Re-consideration of the reaction mechanism after intermediate C to product F is required.
Minor comments:
1) In Figure S40 and S48, arrows for assignment of three products might be pointed to wrong direction. They should be moved to correct positions.
Author Response
- Reviewer comment: “Reaction in HFIP/AcOH (1/1) showed better selectivity for mono-arylation product than the reaction in HFIP/AcOH (3/1) (Table 2, Entry 3). Is it possible to increase the mono-selectivity by optimizing the amount of TDG1 and L1 on the reaction in HFIP/AcOH (1/1)?”
Response: Thanks to the reviewer for this suggestion. To increase the selectivity, we have tried to optimize the amount of TDG1 and L1 in the HFIP/AcOH (1/1) reaction. However, experimental results have showed that the mono-selectivity does not increase.
- Reviewer comment: “Tri-arylation products were obtained in the case of not electron-deficient iodoarene in Scheme 4. Could you provide the reasonable discussion on this trend?”
Response: Thanks to the reviewer for this suggestion. We speculate that the possible reason is that di-arylation products, which bear two electron-deficient arenes, may decrease the reactivity of substrates for their tri-arylation.
- Reviewer comment: “The authors reported DFT-supported reaction mechanism for similar methylene β-C–H arylation of primary aliphatic aldehyde in Sci. 2022, 13, 5938. In this mechanism, hydroxypyridine ligand dissociates prior to oxidative addition of aryl iodide. In contrast, Scheme 6 depicts the oxidative addition of aryl iodide to hydroxypyridine-coordinated palladium(II) complex C. In general, oxidative addition of aryl iodide to palladium complex occurs thorough insertion of palladium to carbon-iodine bond. The previously proposed mechanism should also be applied to the reaction in this paper unless any significant evidence for Scheme 6 in the present form. Furthermore, abstraction of iodo ligand by silver(I) salt prior to reductive elimination to form C–C bond is proposed in the previous paper (Chem. Sci.2022), which conflicts to the mechanism in Scheme 6. Re-consideration of the reaction mechanism after intermediate Cto product F is required.”
Response: Thanks to the reviewer for this suggestion. In the revised manuscript, we have redrawn the mechanism scheme based on our previous paper (Chem. Sci. 2022, 13, 5938). Additionally, the corresponding texts in the mechanism discussion section have also been corrected.
- Reviewer comment: “In Figure S40 and S48, arrows for assignment of three products might be pointed to wrong direction. They should be moved to correct positions.”
Response: Thanks to the reviewer for pointing this out. In the revised Supplementary Materials (Figure S40 and S48), arrows for assignment of three products have been corrected.
Reviewer 3 Report
Comments and Suggestions for Authors
This article describes a continuous work of the lab on β-C(sp3)–H bond arylation of aldehydes. Previous work was on b-methylene C–H arylation of primary aldehydes and the present work describes the β-C(sp3)–H bond arylation of tertiary aliphatic aldehydes by using an α-amino acid as a transient directing group in combination with a 2-pyridone ligand. The optimized reaction condition provides good to moderate yield with various aryl iodides and tertiary aliphatic aldehydes.
I recommend the acceptance of this manuscript to Molecules.
Author Response
Response: The positive feedback provided by the reviewer is greatly appreciated.